# FEW-SHOT LEARNING ON GRAPHS VIA SUPER-CLASSES BASED ON GRAPH SPECTRAL MEASURES

**Jatin Chauhan, Deepak Nathani, Manohar Kaul**
Department of Computer Science
Indian Institute of Technology Hyderabad
{chauhanjatin100,deepakn1019,manohar.kaul}@gmail.com

## ABSTRACT

We propose to study the problem of few-shot graph classification in graph neural networks (GNNs) to recognize unseen classes, given limited labeled graph examples. Despite several interesting GNN variants being proposed recently for node and graph classification tasks, when faced with scarce labeled examples in the few-shot setting, these GNNs exhibit significant loss in classification performance. Here, we present an approach where a probability measure is assigned to each graph based on the spectrum of the graph's normalized Laplacian. This enables us to accordingly cluster the graph base-labels associated with each graph into super-classes, where the $L^p$ Wasserstein distance serves as our underlying distance metric. Subsequently, a super-graph constructed based on the super-classes is then fed to our proposed GNN framework which exploits the latent inter-class relationships made explicit by the super-graph to achieve better class label separation among the graphs. We conduct exhaustive empirical evaluations of our proposed method and show that it outperforms both the adaptation of state-of-the-art graph classification methods to few-shot scenario and our naive baseline GNNs. Additionally, we also extend and study the behavior of our method to semi-supervised and active learning scenarios.

## 1 INTRODUCTION

The need to analyze graph structured data coupled with the ubiquitous nature of graphs (Borgwardt et al., 2005; Duvenaud et al., 2015; Backstrom & Leskovec, 2010; Chau et al., 2011), has given greater impetus to research interest in developing graph neural networks (GNNs) (Defferrard et al., 2016; Kipf & Welling, 2016; Hamilton et al., 2017; Velikovi et al., 2018) for learning tasks on such graphs. The overarching theme in GNNs is for each node's feature vector to be generated by passing, transforming, and recursively aggregating feature information from a given $k$-hop neighborhood surrounding the node. However, GNNs still fall short in the "few-shot" learning setting, where the classifier must generalize well after seeing abundant base-class samples (while training) and very few (or even zero) samples from a novel class (while testing). Given the scarcity and difficulty involved with generation of labeled graph samples, it becomes all the more important to solve the problem of graph classification in the few-shot setting.

**Limitations and challenges:** Recent work by Xu et. al. (Xu et al., 2019) indicated that most recently proposed GNNs were designed based on empirical intuition and heuristic approaches. They studied the representational power of these GNNs and identified that most neighborhood aggregation and graph-pooling schemes had diminished discriminative power. They rectified this problem with the introduction of a novel *injective* neighborhood aggregation scheme, making it as strong as the Weisfeiler-Lehman (WL) graph isomorphism test (Weisfeiler & Leman, 1968).

Nevertheless, the problem posed by extremely scarce novel-class samples in the few-shot setting remains to persist as a formidable challenge, as it requires more rounds of aggregation to affect larger neighborhoods and hence necessitate greater depth in the GNN. However, when it comes to GNNs, experimental studies have shown that an increase in the number of layers results in dramatic performance drops in GNNs (Wu et al., 2019; Li et al., 2018b).

**Our work:** Motivated by the aforementioned observations and challenges, our method does the following. We begin with a once-off preprocessing step. We assign a probability measure to each graph, which we refer to as a *graph spectral measure* (similar to (Gu et al., 2015)), based on the spectrum of the graph's normalized Laplacian matrix representation. Given this metric space of graph spectral measures and the underlying distance as the $L^p$ Wasserstein distance, we compute *Wasserstein barycenters* (Agueh & Carlier, 2011) for each set of graphs specific to a base class and term these barycenters as *prototype graphs*. With this set of prototype graphs for each base class label, we cluster the spectral measures associated with each prototype graph in Wasserstein space to create a super-class label.

Utilizing this super-class information, we then build a *graph of graphs* called a *super-graph*. The intuition behind this is to exploit the non-explicit and latent inter-class relationships between graphs via their spectral measures and use a GNN on this to also introduce a *relational inductive bias* (Battaglia et al., 2018), which in turn affords us an improved sample complexity and hence better combinatorial generalization given such few samples to begin with.

Given, the super-classes and the super-graph, we train our proposed GNN model for few-shot learning on graphs. Our GNN consists of a *graph isomorphism network* (GIN) Xu et al. (2019) as a *feature extractor* $F_\theta(.)$ to generate graph embeddings; on which subsequently acts our *classifier* $C(.)$ comprising of two components: (i) $C^{sup}$: a MLP layer to learn and predict the super class associated to a graph, and (ii) $C^{GAT}$: a *graph attention network* (GAT) to predict the actual class label of a graph. The overall loss function is a sum of the cross-entropy losses associated with $C^{sup}$ and $C^{GAT}$. We follow *initialization based strategy* (Chen et al., 2019), with a training and fine-tuning phase, so that in the fine-tuning phase, the pre-trained parameters associated with $F_\theta(.)$ and $C^{sup}$ are frozen, and the few novel labeled graph samples are used to update the weights and attention learned by $C^{GAT}$.

**Our contributions:** To the best of our knowledge, we are the first to introduce few shot learning on graphs for graph classification. Next, we propose an architecture that makes use of the graph's spectral measures to generate a set of super-classes and a super-graph to better model the latent relations between classes, followed by our GNN trained using an initialization method. Finally, we conduct extensive experiments to gain insight into our method. For example, in the 20-shot setting on the *TRIANGLES* dataset, our method shows a substantial improvement of nearly $7\%$ and $20\%$ over DL-based and unsupervised baselines, respectively.

## 2 RELATED WORK

Few-shot learning in the computer vision community was first introduced by (Fei-Fei et al., 2006) with the intuition that learning the underlying properties of the base classes given abundant samples can help generalize better to unseen classes with few-labeled samples available. Various learning algorithms have been proposed in the *image domain*, among which a broad category of *initialization based methods* aim to learn transferable knowledge from training classes, so that the model can be adapted to unseen classes with limited labeled examples (Finn et al., 2017); (Rusu et al., 2018); (Nichol et al., 2018). Recently proposed and widely accepted *Initialization based methods* can broadly be classified into: (i) methods that learn good model parameters with limited labeled examples and a small number of gradient update steps (Finn et al., 2017) and (ii) methods that learn an optimizer (Ravi & Larochelle, 2017). We refer the interested reader to Chen et. al. (Chen et al., 2019) for more examples of few-shot learning methods in vision.

Graph neural networks (GNNs) were first introduced in (Gori et al., 2005); (Scarselli et al., 2009) as *recurrent message passing algorithms*. Subsequent work (Bruna et al., 2014); (Henaff et al., 2015) proposed to learn smooth spectral multipliers of the graph Laplacian, but incurred higher computational cost. This computational bottleneck was later resolved (Defferrard et al., 2016); (Kipf & Welling, 2016) by learning polynomials of the graph Laplacian. GNNs are a natural extension to Convolutional neural networks (CNNs) on non-Euclidean data. Recent work (Velikovi et al., 2018) introduced the concept of self-attention in GNNs, which allows each node to provide attention to the enclosing neighborhood resulting in improved learning. We refer the reader to (Bronstein et al., 2016) for detailed information on GNNs.

Despite all the success of GNNs, *few-shot classification* remains an under-addressed problem. Some recent attempts have focused on solving the few-shot learning on graph data where GNNs are either trained via co-training and self-training (Li et al., 2018a), or extended by stacking transposed graph convolutional layers imputing a structural regularizer (Zhang et al., 2019) - however, both these works focus only on the node classification task.

To the best of our knowledge, there does not exist any work pertaining few-shot learning on graphs focusing on the graph classification task, thus providing the motivation for this work.

**Comparison to few-shot learning on images:** Few shot learning (FSL) has gained wide-spread traction in the image domain in recent years. However the success of FSL in images is not easily translated to the graph domain for the following reasons: (a) Images are typically represented in Euclidean space and thus can easily be manipulated and handled using well-known metrics like cosine similarity, $L_p$ norms etc. However, graphs come from non-Euclidean domains and exhibit much more complex relationships and interdependency between objects. Furthermore, the notion of a distance between graphs is also not straightforward and requires construction of graph kernels or the use of standard metrics on graph embeddings (Kriege et al., 2019). Additionally, such graph kernels dont capture higher order relations very well. (b) In the FSL setting on images, the number of training samples from various classes is also abundantly more than what is available for graph datasets. The image domain allows training generative models to learn the task distribution and can further be used to generate samples for data augmentation, which act as very good priors. In contrast, graph generative models are still in their infancy and work in very restricted settings. Furthermore, methods like cropping and rotation to improve the models can't be used for graphs given the permutation invariant nature of graphs. Additionally, removal of any component from the graph can adversely affect its structural properties, such as in biological datasets. (c) The image domain has very well-known regularization methods (e.g. Tikhonov, Lasso) that help generalize much better to novel datasets. Although, they dont bring any extra supervised information and hence cannot fully address the problem of FSL in the image domain. To the best of our knowledge, this is still an open research problem in the image domain. On the other hand, in the graph domain, our work would be a first step towards graph classification in an FSL setting, which would then hopefully pave the path for better FSL graph regularizers. (d) Transfer learning has led to substantial improvements on various image related tasks due to the high degree of transferability of feature extractors. Thus, downstream tasks like few-shot learning can be performed well with high quality feature extractor models, such as Resnet variants trained on Imagenet. Transfer learning, or for that matter even good feature extractors, remains a daunting challenge in the graph domain. For graphs, there neither exists a dataset which can serve as a pivot for high quality feature learning, nor does there exist a Graph NN which can capture the higher order relations between various categories of graphs, thus making this a highly challenging problem.

## 3 PRELIMINARIES

In this section, we introduce our notation and provide the necessary background for our few-shot learning setup on graphs. We begin by describing the various data sample types, followed by our learning procedure, in order to formally define few-shot learning on graphs. Finally, we define the *graph spectral distance* between a pair of graphs.

**Data sample sets:** Let $\mathcal{G}$ denote a set of undirected unweighted graphs and $\mathcal{Y}$ be the set of associated class labels. We consider two disjoint populations of labeled graphs consisting of i.i.d. graph samples, the set of *base class* labeled graphs $G_B = \{(g_i^{(B)}, y_i^{(B)})\}_{i=1}^n$ and the set of *novel class* labeled graphs $G_N = \{(g_i^{(N)}, y_i^{(N)})\}_{i=1}^m$, where $g_i^{(B)}, g_i^{(N)} \in \mathcal{G}$, $y_i^{(B)} \in \mathcal{Y}^{(B)}$, and $y_i^{(N)} \in \mathcal{Y}^{(N)}$. Here, the set of base and novel class labels are denoted by $\mathcal{Y}^{(B)} = \{1, \dots, K\}$ and $\mathcal{Y}^{(N)} = \{K+1, \dots, K'\}$, respectively, where $K' > K$. Both $\mathcal{Y}^{(B)}$ and $\mathcal{Y}^{(N)}$ are disjoint subsets of $\mathcal{Y}$, so, $\mathcal{Y}^{(B)} \cap \mathcal{Y}^{(N)} = \emptyset$.

Note that $m \ll n$, i.e., there are far fewer novel class labeled graphs compared to the base class labeled ones. Besides $G_B$ and $G_N$, we consider a set of $t$ unlabeled *unseen* graphs $G_U := \{g_1^{(U)}, \dots, g_t^{(U)} \mid g_i^{(U)} \in \pi_1(G_N), i = 1 \dots t\}$, for testing[1].

---

[1]We use the notation $\pi_1(p)$ and $\pi_2(p)$ to denote the left and right projection of an ordered pair $p$, respectively.

**Learning procedure:** Inspired by the *initialization based methods*, we similarly follow a two-stage approach of *training* followed by *fine-tuning*.

During training, we train a graph feature extractor $F_\theta(G_B)$ with network parameters $\theta$ followed by a classifier $C(G_B)$ on graphs from $G_B$, where the loss function is the standard cross-entropy loss $\mathcal{L}_c$. In order to better recognize and generalize well on samples from novel classes, in the fine-tuning phase, the pre-trained feature extractor $F_\theta(.)$ along with its trained parameters is fixed and the classifier $C(G_N)$ is trained on the novel class labeled graph samples from $G_N$, with the same loss $\mathcal{L}_c$.

Now, given the classification of data samples and the two-stage learning method, our problem of few-shot classification on graphs can be defined as follows.

**Problem definition:** Given $n$ base-class labeled graphs from $G_B$ during the training phase and $m$ novel-class labeled graphs from $G_N$ during the fine-tuning phase, where $m \ll n$, the objective of few-shot graph classification is to classify $t$ unseen test graph samples from $G_U$. Moreover, if $m = qT$, where $T = K' - K$, i.e., each novel class label appears exactly $q$ times in $G_N$, then this setting is referred to as the $q$-shot, $T$-way learning.

**Graph spectral distance:** Let us consider the graphs in $\mathcal{G}$. The normalized Laplacian of a graph $g \in \mathcal{G}$ is defined as $\Delta_g = I - D^{-1/2}AD^{1/2}$, where $A$ and $D$ are the adjacency and the degree matrices of graph $g$, respectively. The set of eigenvalues of $\Delta_g$ given by $\{\lambda_i\}_{i=1}^{|V|}$ is called the *spectrum* of $\Delta_g$ and is denoted by $\sigma(g)$. It is well known that the spectrum $\sigma(g)$ of a normalized Laplacian matrix is contained in interval $[0, 2]$. We assign a Dirac mass $\delta_{\lambda_i}$ concentrated on each $\lambda_i \in \sigma(g)$, thus associating a probability measure to $\sigma(g)$ supported on $[0, 2]$, called the *graph spectral measure* $\mu_{\sigma(g)}$. Furthermore, let $P([0, 2])$ be the set of probability measures on interval $[0, 2]$.

We now define the $p$-th Wasserstein distance between probability measures, which we later use to define the *spectral distance* between a pair of graphs.

**Definition 1** *Let $p \in [1, \infty)$ and let $c : [0, 2] \times [0, 2] \to [0, +\infty]$ be the* cost function *between the probability measures $\mu, \nu \in P([0, 2])$. Then the $p$-th Wasserstein distance between measures $\mu$ and $\nu$ is given by*

$$W_p(\mu, \nu) = \left( \inf_\gamma \int_{[0,2] \times [0,2]} c(x, y)^p d\gamma \mid \gamma \in \Pi(\mu, \nu) \right)^{\frac{1}{p}}$$

*where $\Pi(\mu, \nu)$ is the set of* transport plans*, i.e., the collection of all measures on $[0, 2] \times [0, 2]$ with marginals $\mu$ and $\nu$.*

Given the general definition of the $p$-th Wasserstein distance between probability measures and the *graph spectral measure*, we can now define the spectral distance between a pair of graphs in $\mathcal{G}$.

**Definition 2** *Given two graphs $g, g' \in \mathcal{G}$, the spectral distance between them is defined as*

$$W^p(g, g') := W_p\left( \mu_{\sigma(g)}, \mu_{\sigma(g')} \right)$$

In words, $W^p(g, g')$ is the optimal cost of moving mass from the graph spectral measure of graph $g$ to that of graph $g'$, where the cost of moving unit mass is proportional to the $p$-th power of the difference of real-eigenvalues in interval $[0, 2]^2$.

## 4  OUR METHOD

We present our proposed approach here. First, given abundant base-class labels, we cluster them into *super-classes* by computing *prototype graphs* from each class, followed by clustering the prototype graphs based on their spectral properties. This clustering of prototype graphs induces a natural clustering on their corresponding class labels, resulting in super-classes (as outlined in Section 4.1).

---

[2]In practice, extremely fast computation of $W^p(g, g')$ is achieved using a regularized optimal transport (OT) (Genevay et al., 2016), which makes use of the Sinkhorn algorithm.

Figure 1: The training (left) and fine-tuning (right) stages of our GNN.

These super-classes are then used in the creation of a *super-graph* used further down by our GNN. Note that the creation of super-classes, followed by building a super-graph are a once-off process. The prototype graphs as well as the super-classes for the base classes can be stored in memory for further use.

Next, we explain our graph neural network's architecture which comprises of a *feature extractor* $F_\theta(.)$ and a *classifier* $C(.)$, described in Section 4.2. The classifier $C(.)$ is further subdivided into a classifier $C^{sup}$ that predicts the superclass of a graph feature vector and a *graph attention network* (GAT) $C^{GAT}$ to predict the graph's class label. Figure 1 illustrates the training and fine-tuning phases of our GNN.

## 4.1 Computing Super classes

In order to exploit inter-class relationships between base-class labels, we cluster them in the following manner. First, we partition the set $G_B$ into *class-specific sets* $G^{(i)}$, for $i = 1 \ldots K$, where $G^{(i)}$ is the set of graphs with base-class label $i$. Thus, $G_B = \bigsqcup_{i=1}^{K} G^{(i)}$.

Then, we compute class prototype graphs for each class-specific set. The class prototype graph for class $i$ represented by $p_i$ is given by

$$p_i = \operatorname*{argmin}_{g_i \in \pi_1(G^{(i)})} \frac{1}{|G^{(i)}|} \sum_{j=1}^{|G^{(i)}|} W^p(g_i, g_j) \tag{1}$$

Essentially, the class prototype graph $p_i$ for the $i$-th class is the graph with the least average spectral distance to the rest of the graphs in the same class. Given these $K$ prototypes, we cluster them using Lloyd's method (also known as $k$-means)[3].

**Clustering prototype graphs:** Given $K$ unlabeled prototypes $p_1, \ldots, p_K \in \pi_1(G_B)$ and their associated spectral measures $\mu_{\sigma(p_1)}, \ldots, \mu_{\sigma(p_K)} \in P([0,2])$. We rename the spectral measures as $s_1, \ldots, s_K$ to ease notation. Thus, our goal is to associate these spectral measures to *at most* $k$ clusters, where $k \geq 1$ is a user defined parameter.

The $k$-means problem finds a $k$-partition $C = \{C_1, \ldots, C_k\}$ that minimizes the following objective that represents the overall distortion error of the clustering

$$\operatorname*{argmin}_{C} \sum_{i=1}^{k} \sum_{s_i \in C_i} W_p(s_i, B(C_i)) \tag{2}$$

where $s_i$ is a prototype graph in cluster $C_i$ and $B(C_i)$ is the Wasserstein barycenter of the cluster $C_i$. The barycenter is computed as

$$B(C_i) = \operatorname*{argmin}_{p \in P([0,2])} \sum_{j=1}^{|C_i|} W_p(p, s(i,j)) \tag{3}$$

---

[3]We used the seeding method suggested in $k$-means++ (Arthur & Vassilvitskii, 2007)

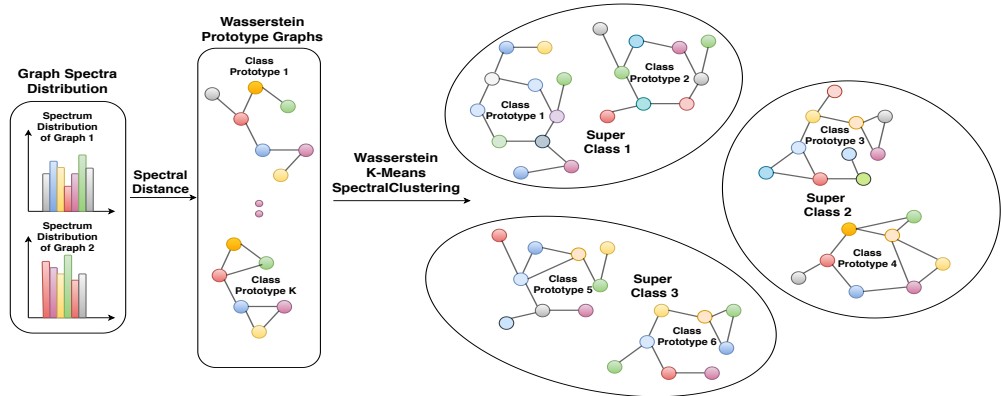

Figure 2: An illustration of our proposed Wasserstein super-class clustering algorithm.

where $s(i, j)$ denotes the $j$-th spectral measure in the $i$-th cluster $C_i$.

**Lloyd's algorithm:** Given an initial set of Wasserstein barycenters $B^{(1)}(C_1), \ldots, B^{(1)}(C_k)$ of spectral measures at step $t = 1$, one uses the standard Lloyd's algorithm to find the solution by alternating between the *assignment* (Equation 4) and *update* (Equation 5) steps

$$C_i^{(t)} = \left\{ s_p : W_p(s_p, B^{(t)}(C_i)) \leq W_p(s_p, B^{(t)}(C_j)), \forall j, 1 \leq j \leq k, 1 \leq p \leq K \right\} \quad (4)$$

$$C_i^{(t+1)} = B(C_i^{(t)}) \quad (5)$$

Lloyd's algorithm is known to converge to a local minimum (except in pathological cases, where it can oscillate between equivalent solutions). The final output is a grouping of the prototype graphs into $k$ groups, which also induces a grouping of the corresponding base classes. We denote these class groups as super-classes and denote the set of super-classes as $\mathcal{Y}^{sup}$.

## 4.2 OUR GRAPH NEURAL NETWORK

**Feature extractor:** To apply standard neural network architectures for downstream tasks we must embed the graphs in a finite dimensional vector space. We consider graph neural networks (GNNs) that employ the following *message-passing* architecture

$$H^{(j)} = M(A, H^{(j-1)}, \theta^{(j)})$$

where $H^{(j)} \in \mathbb{R}^{|V| \times d}$ are the node embeddings (i.e., messages) computed after $j$ steps of the GNN and $M$ is the message propagation function which depends on the adjacency matrix of the graph $A$, the trainable parameters of the $j^{th}$ layer $\theta^{(j)}$, and node embeddings $H^{(j-1)}$ generated from the previous step.

A recently proposed GNN called the *graph isomorphism network* (GIN) by Xu et al. (2019) was shown to be stronger than several popular GNN variants like GCN Kipf & Welling (2016) and GraphSAGE Hamilton et al. (2017). What makes GIN so powerful and sets it apart from the other GNN variants is its *injective* neighborhood aggregation scheme which allows it to be as powerful as the Weisfeiler-Lehman (WL) graph isomorphism test. Motivated by this finding, we chose GIN as our graph feature extractor. The message propagation scheme in GIN is given by

$$H^{(j)} = MLP((1 + \epsilon)^j \odot H^{(j-1)} + A^T H^{(j-1)}) \quad (6)$$

Here, $\epsilon$ is a layer-wise learnable scalar parameter and $MLP$ represents a multi-layer perceptron with layer-wise non-linearities for more expressive representations. The full GIN model run $R$ iterations of Equation 6 to generate final node embeddings which we represent by $H^{(R)}$. As features from earlier iterations can also be helpful in achieving higher discriminative power, embeddings $H^{(j)}$

from all $R$ iterations are concatenated as

$$H_g = \left\|_{j=1}^{R} H^{(j)} \right.,$$

Here, $H^{(j)} = \sum_{v \in V} H_v^{(j)}$, where $H_i^{(j)}$ represents the $i$-th node's embedding in the $j$-th iteration and $\|$ denotes a concatenation operator. $H_g$ now contains the graph embedding of a graph $g$ and is passed on to the classifier.

**Classifier:** Here, our objective is to improve the class separation produced by the graph embeddings of the feature extractor $F_\theta(.)$ and we do this by building a "graph of graph embeddings", called a *super-graph* $g^{sup}$, where each node is a graph feature vector. We then employ our classifier $C(.)$ on this super-graph to achieve better separation among the graph classes in the embedding space.

During training, we first build the super-graph $g^{sup}$ on a batch of base-labeled graphs as a *collection* of $k$-NN graphs, where each constituent $k$-NN graph is built on the graphs belonging to the same super-class. $g^{sup}$ is then passed through a multi-layered graph attention network $C^{GAT}$ to learn the associated class probabilities. The features extracted from $F_\theta(.)$ are passed into the MLP network $C^{sup}$ to learn the associated super-class labels. $C^{sup}$ and $C^{GAT}$ combine to form our classifier $C(.)$. The cross-entropy losses associated with $C^{sup}$ and $C^{GAT}$ are added to give the overall loss for $C(.)$. The intuition behind the construction of $g^{sup}$ to train $C^{GAT}$ on was to further improve the existing cluster separation based on graph spectral measures by introducing a *relational inductive bias* (Battaglia et al., 2018) that is inherent to the GNN $C^{GAT}$.

Recall that we adopt an *initialization method* (described in 3). In our fine-tuning stage, novel class labeled graphs from $G_N$ are input to the network. The pre-trained parameters learned by the feature extractor $F_\theta(.)$ are *fixed* and $C^{sup}$ is used to infer the novel graph's super-class label, followed by creation of super-graph on the novel graph samples and finally updating the parameters in $C^{GAT}$ through the loss. Finally the evaluation is performed on the samples from the unseen test set $G_U$.

**Discussion:** We make the assumption that the novel test classes belong to the same set of super-classes from the training graphs. The reason being that the novel class labeled samples are so much fewer than the base class labeled samples, that the resulting super-graph ends up being extremely sparse and deviates a lot from the shape of the super-graph from the base classes; therefore it severely hinders $C^{GAT}$s ability to effectively aggregate information from the embeddings of the novel class labeled graphs. Instead, we pass the novel graph samples through our trained $C^{sup}$ and infer its super-class label and this works very effectively for us, as is evidenced by our empirical results.

## 5 EXPERIMENTAL RESULTS

### 5.1 BASELINES AND DATASETS

The standard graph classification datasets do not adequately satisfy the requirements for few-shot learning due to the dearth of unique class labels. Hence, we pick four new classification datasets, namely, *Letter-High*, *TRIANGLES*, *Reddit-12K*, and *ENZYMES*. The details and statistics for these datasets are given in Appendix A.1. As there do not exist any standard state-of-the-art methods for few-shot graph classification, we chose existing baselines for standard graph classification from both supervised and unsupervised methods.

For *supervised deep learning* baselines, we chose - GIN (Xu et al. (2019)), CapsGNN (Xinyi & Chen (2019)), and Diffpool (Lee et al. (2019)). We ran these methods with similar settings as ours, i.e., by partitioning the main model into *feature extraction* and *classifier* sub-models to compare them in a fair and informative manner. From the *unsupervised* category, we consider 4 powerful SOTA methods - AWE (Ivanov & Burnaev (2018)), Graph2Vec (Narayanan et al. (2017)), Weisfeiler-Lehman subtree Kernel (Shervashidze et al. (2011)), and Graphlet count kernel (Shervashidze et al. (2009)). Since we want to analyze the few-shot classification abilities of these models, we essentially want to find out how well these algorithms can achieve class separation. We use $k$-NN search on the output embeddings of these algorithms.

Table 1: Results for various few-shot scenarios on *Letter-High* and *TRIANGLES* datasets. The best results are highlighted in **bold** while the second best results are underlined.

| Method | *Letter-High* | | | *TRIANGLES* | | |
|---|---|---|---|---|---|---|
| | 5-shot | 10-shot | 20-shot | 5-shot | 10-shot | 20-shot |
| *WL* | $65.27 \pm 7.67$ | $68.39 \pm 4.69$ | $72.69 \pm 3.02$ | $51.25 \pm 4.02$ | $53.26 \pm 2.95$ | $57.74 \pm 2.88$ |
| *Graphlet* | $33.76 \pm 6.94$ | $37.59 \pm 4.60$ | $41.11 \pm 3.71$ | $40.17 \pm 3.18$ | $43.76 \pm 3.09$ | $45.90 \pm 2.65$ |
| *AWE* | $40.60 \pm 3.91$ | $42.20 \pm 2.87$ | $43.12 \pm 1.00$ | $39.36 \pm 3.85$ | $42.58 \pm 3.11$ | $44.98 \pm 1.54$ |
| *Graph2Vec* | $66.12 \pm 5.21$ | $68.17 \pm 4.26$ | $70.28 \pm 2.81$ | $48.38 \pm 3.85$ | $50.16 \pm 4.15$ | $54.90 \pm 3.01$ |
| *Diffpool* | $58.69 \pm 6.39$ | $61.59 \pm 5.21$ | $64.67 \pm 3.21$ | $64.17 \pm 5.87$ | $67.12 \pm 4.29$ | $73.27 \pm 3.29$ |
| *CapsGNN* | $56.60 \pm 7.86$ | $60.67 \pm 5.24$ | $63.97 \pm 3.69$ | $65.40 \pm 6.13$ | $68.37 \pm 3.67$ | $73.06 \pm 3.64$ |
| *GIN* | $65.83 \pm 7.17$ | $69.16 \pm 5.14$ | $73.28 \pm 2.17$ | $63.80 \pm 5.61$ | $67.30 \pm 4.35$ | $72.55 \pm 1.97$ |
| *GIN-$k$-NN* | $63.52 \pm 7.27$ | $65.66 \pm 8.69$ | $67.45 \pm 8.76$ | $58.34 \pm 3.91$ | $61.55 \pm 3.19$ | $63.45 \pm 2.76$ |
| *OurMethod-GCN* | $68.69 \pm 6.50$ | $72.80 \pm 4.12$ | $75.17 \pm 3.11$ | $69.37 \pm 4.92$ | $73.11 \pm 3.94$ | $77.86 \pm 2.84$ |
| **OurMethod-GAT** | $\mathbf{69.91 \pm 5.90}$ | $\mathbf{73.28 \pm 3.46}$ | $\mathbf{77.38 \pm 1.58}$ | $\mathbf{71.40 \pm 4.34}$ | $\mathbf{75.60 \pm 3.67}$ | $\mathbf{80.04 \pm 2.20}$ |

Further configuration and implementation details for the baselines can be found in Appendix A.2. We also emphasize the benefit of using a GNN as a classifier by showing the adaptation of our model to semi-supervised fine-tuning (in Appendix A.5) and active learning (in Appendix A.6) settings.

## 5.2 FEW-SHOT RESULTS

We consider two variants of our model as *naive baselines*. In the first variant, we replace our GAT classifier with GCN Kipf & Welling (2016). We call this model *OurMethod-GCN*. This variant is used to justify the choice of GAT over GCN.

In the second variant, we replace the entire classifier with the $k$-NN algorithm over the features extracted from various layers of the *feature extractor*. We call this variant *GIN-$k$-NN* and this is introduced to emphasize the significance of building a super-graph and using a GAT on it as a classifier to exploit the relational inductive bias.

The results for all the datasets in various $q$-shot scenarios, where $q \in \{5, 10, 20\}$ are given in Table 1. We run each model 50 times and report averaged results. In every run, we select a different novel labeled subset $G_N$ for fine-tuning the classifiers of the models. The evaluation for all models is done by randomly selecting a subset of 500 samples from the testing set $G_U$ for *Letter-High* and *TRIANGLES*, whereas over 150 for *ENZYMES* and 300 for *Reddit* dataset and averaging over 10 such random selections.

Table 2: Results for various few-shot scenarios on *Reddit-12K* and *ENZYMES* datasets. The best results are highlighted in **bold** while the second best results are underlined.

| Method | *Reddit-12K* | | | *ENZYMES* | | |
|---|---|---|---|---|---|---|
| | 5-shot | 10-shot | 20-shot | 5-shot | 10-shot | 20-shot |
| *WL* | $40.26 \pm 5.17$ | $42.57 \pm 3.69$ | $44.41 \pm 3.43$ | $55.78 \pm 4.72$ | $58.47 \pm 3.84$ | $60.1 \pm 3.18$ |
| *Graphlet* | $33.76 \pm 6.94$ | $37.59 \pm 4.60$ | $41.11 \pm 3.71$ | $53.17 \pm 5.92$ | $55.30 \pm 3.78$ | $56.90 \pm 3.79$ |
| *AWE* | $30.24 \pm 2.34$ | $33.44 \pm 2.04$ | $36.13 \pm 1.89$ | $43.75 \pm 1.85$ | $45.58 \pm 2.11$ | $49.98 \pm 1.54$ |
| *Graph2Vec* | $27.85 \pm 4.21$ | $29.97 \pm 3.17$ | $32.75 \pm 2.02$ | $55.88 \pm 4.86$ | $58.22 \pm 4.30$ | $62.28 \pm 4.14$ |
| *Diffpool* | $35.24 \pm 5.69$ | $37.43 \pm 3.94$ | $39.11 \pm 3.52$ | $45.64 \pm 4.56$ | $49.64 \pm 4.23$ | $54.27 \pm 3.94$ |
| *CapsGNN* | $36.58 \pm 4.28$ | $39.16 \pm 3.73$ | $41.27 \pm 3.12$ | $52.67 \pm 5.51$ | $55.31 \pm 4.23$ | $59.34 \pm 4.02$ |
| *GIN* | $40.36 \pm 4.69$ | $43.70 \pm 3.98$ | $46.28 \pm 3.49$ | $55.73 \pm 5.80$ | $58.83 \pm 5.32$ | $61.12 \pm 4.64$ |
| *GIN-$k$-NN* | $41.31 \pm 2.84$ | $43.58 \pm 2.80$ | $45.12 \pm 2.19$ | $\mathbf{57.24 \pm 7.06}$ | $59.34 \pm 5.24$ | $60.49 \pm 3.48$ |
| *OurMethod-GCN* | $40.77 \pm 4.32$ | $44.28 \pm 3.86$ | $48.67 \pm 4.22$ | $54.34 \pm 5.64$ | $58.16 \pm 4.39$ | $60.86 \pm 3.74$ |
| **OurMethod-GAT** | $\mathbf{41.59 \pm 4.12}$ | $\mathbf{45.67 \pm 3.68}$ | $\mathbf{50.34 \pm 2.71}$ | $55.42 \pm 5.74$ | $\mathbf{60.64 \pm 3.84}$ | $\mathbf{62.81 \pm 3.56}$ |

The results clearly show that our proposed method and its GCN variant (i.e., *OurMethod-GCN*) outperform the baselines. GIN-$k$-NN shows significant degradation in results on nearly all the three paradigms for all datasets with exceptions on 5-shot and 10-shot on *ENZYMES* dataset, thus strongly indicating that the improvements of our method can primarily be attributed to our GNN classifier fed with the super-graph constructed from our proposed method. The improvements in results are higher on *TRIANGLES* and *Reddit* datasets in contrast to *Letter-High* and *ENZYMES*, which can be attributed to the smaller size of the graphs in *Letter-High* making it difficult to distinguish based on graph spectra alone, whereas the complex and highly inter-related structure of enzymes makes it dif-

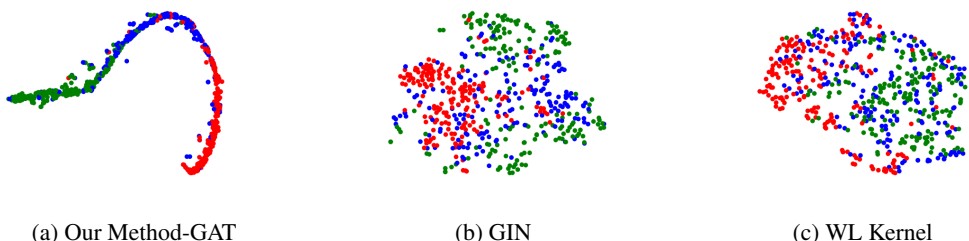

| (a) Our Method-GAT | (b) GIN | (c) WL Kernel |

Figure 3: Visualization: t-SNE plots of the computed embeddings of test graphs on 20-shot scenario from OurMethod-GAT (left), GIN (middle) and WL Kernel (right) on *TRIANGLES* dataset. The embeddings for both our model and GIN are taken from the final layers of the respective models.

ficult for the DL based feature extractors as well as the graph kernel methods to segregate the classes in the feature and graph space respectively. GIN and WL show much better results as compared to other baselines for all the $q$-shot scenarios, whereas AWE and Graphlet Kernel show significantly low results, unable to capture the properties of the graphs well. The DL baselines apart from GIN on the other hand show improvements on the *TRIANGLES* dataset performing close to GIN, where the unsupervised methods fails to capture the local node properties, however still perform poorly on other datasets. For the 20-shot scenario on *TRIANGLES*, our GAT variant shows an improvement of around 7% over DL baselines and more than 20% when compared to unsupervised methods. The substantial improvements of around 4% on *Reddit* dataset shows the superiority of our model for both the variants - GAT and GCN. Furthermore, the $t$-SNE plots in Figure 3 show a substantial and interesting separation of class labels which strongly indicate that a good feature extractor in conjunction with a GNN perform well as a combination. The t-SNE plots for *ENZYMES*, *Reddit*, and *Letter-High* are shown in figures 4, 5 and 6 respectively.

## 5.3 ABLATION STUDY ON NUMBER OF SUPER-CLASSES

Here, we study the behavior of our proposed network model without the super-class classifier $C^{sup}$. In Table 3 (10 and 20-shot setting), we observe a marked increase with the addition of our classifier which uses the super-class information and the super-graph based on spectral measures to guide $C^{GAT}$ towards improving the class separation of the graphs during both the training and fine-tuning stages. Using super-classes help in reducing the sample complexity of the large Hypothesis space and makes tuning of the model parameters easier during fine-tuning stage with less samples and few iterations. Negligible differences are observed on *ENZYMES* dataset since both the number of training classes as well as test classes are low, thus, the model performs equally well on removing super-classes. This is because of the latent inter-class representations can still be captured between few classes especially during the fine-tuning phase.

Table 3: Ablation Study: "No-SC" represents our classifier $C(.)$ without $C^{sup}$ and "With-SC" represents $C(.)$ with both $C^{sup}$ and $C^{GAT}$ present.

| Dataset | 10-shot | | 20-shot | |
|---|---|---|---|---|
| | No-SC | With-SC | No-SC | With-SC |
| *Letter-High* | $71.13 \pm 3.64$ | $73.61 \pm 3.19$ | $75.23 \pm 2.48$ | $77.42 \pm 1.47$ |
| *TRIANGLES* | $74.03 \pm 3.89$ | $76.49 \pm 3.26$ | $76.89 \pm 2.63$ | $80.14 \pm 1.88$ |
| *Reddit-12K* | $43.76 \pm 4.34$ | $45.35 \pm 4.06$ | $48.19 \pm 4.01$ | $50.36 \pm 3.04$ |
| *ENZYMES* | $59.97 \pm 3.98$ | $59.58 \pm 4.32$ | $62.7 \pm 3.63$ | $62.39 \pm 3.48$ |

## 5.4 SENSITIVITY ANALYSIS OF VARIOUS ATTRIBUTES

Our proposed method contains two crucial attributes. We analyze our model by varying: (i) the number of super-classes and (ii) the $k$-value in super-graph construction. The effect of varying these attributes on model accuracy are shown in Tables 4 and 5, respectively. For *TRIANGLES* and *Letter-*

Table 4: Model analysis over number of super-classes in 20-shot scenario. There is no evaluation for 5 super-classes on ENZYMES since the number of training classes is 4. Default value of parameter $k$ is fixed at 2.

| Dataset | 20-shot | | | | |
|---|---|---|---|---|---|
| . | 1 | 2 | 3 | 4 | 5 |
| *Letter-High* | $74.43 \pm 2.61$ | $76.61 \pm 1.67$ | $77.51 \pm 1.49$ | $76.31 \pm 1.98$ | $75.05 \pm 2.29$ |
| *TRIANGLES* | $76.43 \pm 2.87$ | $79.55 \pm 1.91$ | $80.51 \pm 1.72$ | $78.91 \pm 2.09$ | $78.25 \pm 2.40$ |
| *Reddit-12K* | $48.32 \pm 4.09$ | $50.67 \pm 2.94$ | $50.10 \pm 3.02$ | $49.52 \pm 4.02$ | $48.33 \pm 4.08$ |
| *ENZYMES* | $62.34 \pm 4.11$ | $62.13 \pm 4.01$ | $60.16 \pm 3.81$ | $59.34 \pm 3.98$ | - |

Table 5: Model analysis over number of neighbors ($k$) in super-graph for 20-shot scenario. Default value for the number of super-classes is fixed at 3.

| Dataset | 20-shot | | | | |
|---|---|---|---|---|---|
| | 2 | 4 | 6 | 8 | Heuristic |
| *Letter-High* | $77.33 \pm 1.71$ | $76.61 \pm 1.67$ | $75.63 \pm 2.49$ | $74.66 \pm 2.61$ | $74.35 \pm 2.48$ |
| *TRIANGLES* | $80.77 \pm 1.57$ | $79.85 \pm 1.59$ | $79.45 \pm 1.97$ | $78.93 \pm 2.04$ | $79.42 \pm 3.16$ |
| *Reddit-12K* | $50.48 \pm 3.02$ | $46.37 \pm 3.03$ | $44.12 \pm 2.98$ | $43.88 \pm 3.24$ | $44.82 \pm 2.83$ |
| *ENZYMES* | $62.34 \pm 4.11$ | $61.42 \pm 4.42$ | $60.23 \pm 5.10$ | $59.67 \pm 4.77$ | $61.07 \pm 4.68$ |

*High* datasets, as we increase the number of super-classes, we observe the accuracy improving steadily up to 3 super-classes and then dropping from there onwards. For super-classes less than 3, we observe that the $k$-NN graph does not respect the class boundaries that are already imposed by the graph spectral measures, thus connecting more arbitrary classes. For *Reddit* we observe the performance is slightly better on using 2 super-classes and for *ENZYMES* similar performances are observed for both 1 and 2 super-classes as described in the ablation study. On the other hand, increasing the number of super-classes past 3, makes each super-class cluster very sparse with few graph classes within, leading to an underflow of information between the graph classes. The same effect is observed for all datasets.

The $k$-value or the number of neighbors of each node belonging to the same *connected component* in the super-graph (i.e., belonging to the same super-class) is another salient parameter upon which hinges the information flow (via message passing) between the graphs of the same super-class. We analyze our model with $k$ values in the set $\{2, 4, 6, 8\}$ and a commonly used heuristic method, whereby each graph is connected to $\sqrt{b_s}$ nearest neighboring graphs based on the Euclidean similarity of their feature representations, where $b_s$ is the number of samples in the mini-batch corresponding to super-classes $s$. We achieve best results with 2-NN graphs per super-class and increasing $k$ beyond it leads to denser graphs with unnecessary connections between classes belonging to the same super-class.

## 6 CONCLUSION

In this paper, we investigated the problem of few-shot learning on graphs for the graph classification task. We explicitly created a *super-graph* on the base-labeled graphs and then *grouped / clustered* their associated class labels into *super-classes*, based on the graph spectral measures attributed to each graph and the $L^p$-Wasserstein distances between them. We found that training our GNN on the super-graph along with the auxiliary super-classes resulted in a marked improvement over state-of-the-art GNNs. A promising future work is to propose new GNN models that break away from current neighborhood aggregation schemes to specifically overcome the obstacle posed by few-shot learning on graphs. Our source-code and dataset splits have been made public in an attempt to attract more attention to the context of few-shot learning on graphs.

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

# A   Appendix

## A.1   Dataset details

We use 4 different datasets namely - *Reddit-12K*, *ENZYMES*, *Letter-High* and *TRIANGLES* to perform exhaustive empirical evaluation of our model on various real-world datasets varying from small average graph size on *Letter-High* to large graphs like *Reddit-12K*. These datasets can be downloaded here [4]. The dataset statistics are provided in Table 6, while the split statistics are provided in Table 7

Table 6: Dataset Statistics

| Dataset Name | # Classes | # Graphs | Avg # Nodes | Avg # Edges |
|---|---|---|---|---|
| *Reddit-12K* | 11 | 11929 | 391.41 | 456.89 |
| *ENZYMES* | 6 | 600 | 32.63 | 62.14 |
| *Letter-High* | 15 | 2250 | 4.67 | 4.50 |
| *TRIANGLES* | 10 | 45000 | 20.85 | 35.50 |

**Dataset Description: Reddit-12K** datasets contains 11929 graphs where each graph corresponds to a thread in which each node represents a user and each edge represents that one user has responded to a comment from some other user. There are 11 different types of discussion forums corresponding to each of the 11 classes.
**ENZYMES** is a dataset of protein tertiary structures consisting of 600 enzymes from the BRENDA enzyme database. The dataset contains 6 different graph categories corresponding to each different top-level EC enzyme.
**TRIANGLES** dataset contain 10 different classes where the classes are numbered from 1 to 10 corresponding to the number of triangles/3-cliques in each graph of the dataset.
**Letter-High** dataset contains graphs which represent distorted letter drawings from the english alphabets - $A, E, F, H, I, K, L, M, N, T, V, W, X, Y, Z$. Each graph is a prototype manual construction of the alphabets.

Table 7: Dataset Splits

| Dataset Name | # Train Classes | # Test Classes | # Training Graphs | # Validation Graphs | # Test Graphs |
|---|---|---|---|---|---|
| *Reddit-12K* | 7 | 4 | 566 | 141 | 404 |
| *ENZYMES* | 4 | 2 | 320 | 80 | 200 |
| *Letter-High* | 11 | 4 | 1330 | 320 | 600 |
| *TRIANGLES* | 7 | 3 | 1126 | 271 | 603 |

The validation graphs are used to assess model performance on training classes itself to check overfitting as well as for grid-search over hyperparameters. The actual train-testing class splits used for this paper are provided with the code. Since the TRIANGLES dataset has a large number of samples, this makes it infeasible to run many baselines including DL and non-DL methods. Hence, we sample 200 graphs from each class, making the total sample size 2000. Similarly we downsample the number of graphs from 11929 to 1111 (nearly 101 graphs per class). Downsampling is performed for Reddit-12K given extremely large graph sizes which makes the graph kernels as well as some deep learning baselines extremely slow.

---

[4]https://ls11-www.cs.tu-dortmund.de/staff/morris/graphkerneldatasets

## A.2 BASELINE DETAILS

This section details the implementation of the baseline methods. Since, DL-based methods - GIN, CapsGNN and DIFFPOOL have not been previously run on these datasets, we select the crucial hyper-parameters - such as number of layers heuristically based on the results of standard graph classification datasets on the best performing variants of these models. For these three methods we take the novel layers proposed in the corresponding papers as their feature extractors, while downstream MLP layers are chosen as the classifier. The training and evaluation strategies are similar to our model, i.e., the models are first trained in an end-to-end fashion on the training dataset $G_B$ until convergence with learning rate decay on loss plateau and then the classifier layers are fine-tuned over $G_N$, keeping the parameters of the feature extractor layers fixed.

For the unsupervised models - WL subtree kernel, Graphlet Count kernel, AWE and Graph2Vec, the evaluation is done using $k$-NN search to assess the clustering quality of these models in our few-shot scenario. We refrain from using high-level classifier models such as SVM or MLPs, since training these classifiers on few-shot regime will not properly assess the abilities of these models to cluster together graphs of similar class labels. We empirically found that using high level classifiers resulted in higher deviations and lower mean accuracies. We choose the hyper-parameters for these models using grid-search, since they are significantly faster and each one of these models have few highly sensitive parameters which affect the model significantly. For these models, we perform a grid search for selection of $k$ in the $k$-NN algorithm from the set $\{1, 2, 3, 4, 5\}$ for the 5-shot scenario, of which $k = 1$ was found to perform the best. For higher shot scenario, the search was performed over the set $\{1, 2, 3, 4, 5, 6, 7, 8, 9, 10\}$, where we again found $k = 1$ to be the best. The validation set is used to check overfitting and hyper-parameter selection on the baseline methods.

## A.3 OUR MODEL DETAILS

This section provides the implementation details of our proposed model. Since, our feature extractor model is GIN, we maintain similar parameter settings as recommended by their paper. As mentioned in section 4.2, using embeddings from all iterations of the message passing network helps achieve better discriminative power and improved gradient flow, therefore we employ the same strategy in our feature extractor. The number of super-classes are selected from the set $\{1, 2, 3, 4, 5\}$ using grid-search. The $k$-value for construction of super-graph was selected from the set $\{2, 4, 6, 8\}$. The feature extractor model uses *batch-normalization* between subsequent message passing layers. We use dropout of 0.5 in the $C^{sup}$ layers. The $C^{GAT}$ layers undergo normalization of inputs between subsequent layers along with a dropout of 0.5, however, the normalization mechanism in classifier layers is different from batch-norm. We normalize each feature embedding to have Euclidean norm with value 1. Essentially,

$$\mathbf{x}_{input}^{j+1} = \frac{\mathbf{x}_{out}^{j}}{||\mathbf{x}_{out}^{j}||_2} \tag{7}$$

where $\mathbf{x}_{input}^{j+1}$ is the input of $j + 1^{th}$ layer of classifier, $\mathbf{x}_{out}^{j}$ is the output of the $j^{th}$ layer. The inputs of the first layer of $C^{GAT}$ also undergo the same transformation over the outputs of the feature extractor model. We train our models with Adam (Kingma & Ba (2014)) with an initial learning rate of $10^{-3}$ for 50 epochs. Each epoch has 10 iterations, where we randomly select a mini-batch from the training data $G_B$. The fine-tuning stage consists of 20 epochs with 10 iterations per epoch. We use a two-layer MLP over the final attention layer of $C^{GAT}$ for classification. The attention layers use multi-head attention with 2 heads and leaky ReLU slope of 0.1 . The embeddings from both the attention heads are concatenated. For 20-shot, we set $k$ to 2, number of super-classes to 3 and batch size to 128 on the *Letter-High* dataset, while $k$ is set to 2 and batch size 64 on *Reddit*, *ENZYMES* and *TRIANGLES* datasets. The number of super-classes for *Reddit* are set to 2, for *ENZYMES* it is set to 1 and for *TRIANGLES* are 3. For *ENZYMES*, there are negligible differences on using 1 and 2 super-classes as shown in table 4. We used *Python Optimal Transport* (POT) library [5] for implementation of the $p$-th Wasserstein distance.

---

[5]https://pot.readthedocs.io/en/stable/all.html

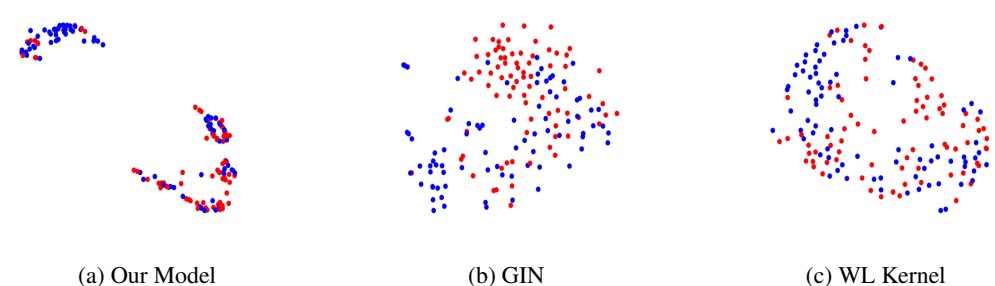

(a) Our Model          (b) GIN          (c) WL Kernel

Figure 4: Visualization: t-SNE plots of the computed embeddings of test graphs on 20-shot scenario from OurMethod-GAT (left), GIN (middle) and WL Kernel (right) on *ENZYMES* dataset.

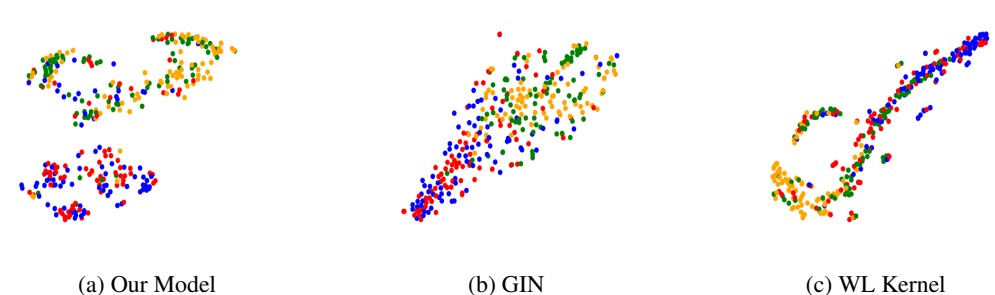

(a) Our Model          (b) GIN          (c) WL Kernel

Figure 5: Visualization: t-SNE plots of the computed embeddings of test graphs on 20-shot scenario from OurMethod-GAT (left), GIN (middle) and WL Kernel (right) on *Reddit* dataset.

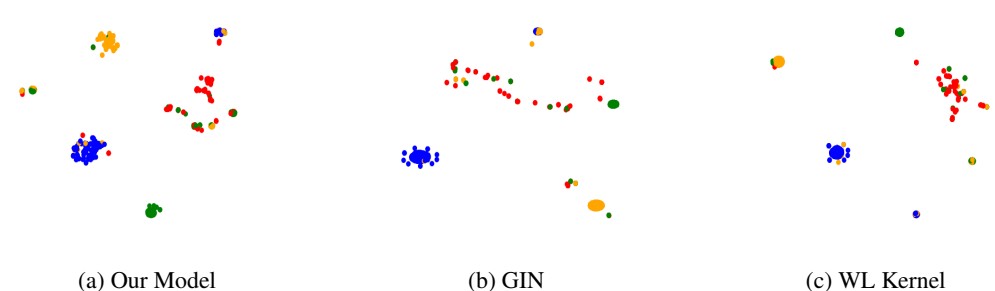

(a) Our Model          (b) GIN          (c) WL Kernel

Figure 6: Visualization: t-SNE plots of the computed embeddings of test graphs on 20-shot scenario from OurMethod-GAT (left), GIN (middle) and WL Kernel (right) on *Letter-High* dataset.

## A.4 SILHOUETTE SCORES

To assess the clustering abilities of the models we analyze the *silhouette scores* of the test embeddings produced by the GAT variant of our method, GIN and WL Kernel. Silhouette coefficient essentially measures the ratio of intra-class versus inter-class distance. The Silhouette Coefficient is calculated using the mean intra-cluster distance (a) and the mean nearest-cluster distance (b) for each sample. The Silhouette Coefficient for a sample is given by $\frac{(b-a)}{max(a,b)}$ ,where $b$ is the distance between a sample and the nearest cluster that the sample is not a part of. The results for mean silhouette coefficient over the test samples averaged over multiple runs are shown in Table 8. We normalize the embeddings before calculating the silhouette coefficient. We can clearly see that our model creates better clusters with low intra-cluster distance as well as high inter-cluster distance. Note that the coefficient value for WL remains the same for all scenarios since it computes fixed embeddings attributed to absence of any DL component.

Table 8: Silhouette coefficients of the test classes for the three dominant models - GAT variant of Our Method, GIN and WL. The best scores are highlighted in bold.

| Method | Reddit-12K | | ENZYMES | | Letter-High | | TRIANGLES | |
|---|---|---|---|---|---|---|---|---|
| | 10-shot | 20-shot | 10-shot | 20-shot | 10-shot | 20-shot | 10-shot | 20-shot |
| GIN | -0.0566 | -0.0652 | 0.0168 | 0.0432 | 0.2157 | 0.2316 | 0.0373 | 0.1256 |
| WL Kernel | -0.0626 | -0.0626 | **0.0366** | 0.0366 | 0.2490 | 0.2490 | 0.0186 | 0.0186 |
| OurMethod-GAT | **-0.0553** | **-0.0559** | 0.0296 | **0.1172** | **0.3494** | **0.3787** | **0.3824** | **0.4508** |

Table 9: Semi-supervised fine-tuning results for various $p$ values on 10-shot and 20-shot scenarios, where "No Semi-Sup" represents the fine-tuning stage without additional labeled samples.

| Dataset | 10-shot | | | 20-shot | | |
|---|---|---|---|---|---|---|
| | No Semi-Sup | 25 | 50 | No Semi-Sup | 25 | 50 |
| Letter-High | $73.21 \pm 3.19$ | $74.18 \pm 2.58$ | $74.65 \pm 2.16$ | $76.95 \pm 1.79$ | $77.79 \pm 1.52$ | $78.31 \pm 1.11$ |
| TRIANGLES | $75.83 \pm 2.97$ | $76.36 \pm 2.59$ | $77.8 \pm 2.04$ | $80.09 \pm 1.78$ | $81.29 \pm 1.98$ | $81.87 \pm 1.45$ |

| Dataset | 10-shot | | | 20-shot | | |
|---|---|---|---|---|---|---|
| | No Semi-Sup | 10 | 20 | No Semi-Sup | 10 | 20 |
| Reddit | $45.41 \pm 3.79$ | $45.88 \pm 3.32$ | $46.01 \pm 2.99$ | $50.34 \pm 2.77$ | $50.76 \pm 2.52$ | $51.17 \pm 2.21$ |
| ENZYMES | $60.13 \pm 3.98$ | $60.87 \pm 3.24$ | $61.25 \pm 3.17$ | $62.74 \pm 3.64$ | $63.10 \pm 3.47$ | $63.67 \pm 3.18$ |

### A.5 SEMI-SUPERVISED FINE-TUNING

In many real-world learning scenarios, it is quite common to find abundant unlabelled data. Since our model uses a GNN classifier, this makes it possible to use unlabelled data while learning through message passing, where the fine tuning stage of our method is performed in semi-supervised settings.

Essentially, while fine tuning the model, i.e., only training the classifier $C^{GAT}$ on $G_N$, we additionally use $p$ more graphs along with $G_N$, whose labels are unknown. The learning objective for fine tuning stage doesn't change since the gradients are back-propagated from the labeled samples only. In this setting, each node in the attention classifier can aggregate information from unlabelled samples as well, thus allowing improved learning of the graphs features in $C^{GAT}$. We show the results for $p$ values 25 and 50 on *Letter-High* and *TRIANGLES* datasets, whereas for $p$ values 10 and 20 on *Reddit* and *ENZYMES* datasets. The results are shown in Table 9. We observe an increase in the accuracy with increase in number of unlabeled samples during fine-tuning phase.

### A.6 ADAPTATION TO ACTIVE-LEARNING

In this section, we show the adaptation of our model to highly practical *active learning* scenario. In many real world applications, we might start with few samples per class, however as the number of samples to classify from these classes increase over time, some of these samples can be used by the model to adaptively learn and improve with very less human intervention, since the number of number of samples to be queried for theirs label can always be controlled.

To perform active-learning, we first select a random subset of size 100 for *Letter-High* and *TRIANGLES* datasets as well as a random subset of size 40 for *Reddit* and *ENZYMES* datasets, which we term as $G_{random}$, then fine tune the model on $G_N$ and further evaluate the model on $G_{random}$.

Table 10: Active Learning Results. The value below each shot represents the number samples $l$, added to $G_N$ for second fine-tuning step, where "No AL" represents the model evaluation without additional labeled samples.

| Dataset | 10-shot | | | 20-shot | | |
|---|---|---|---|---|---|---|
| | No AL | 15 | 25 | No AL | 15 | 25 |
| Letter-High | $73.34 \pm 3.37$ | $75.03 \pm 3.24$ | $76.89 \pm 2.16$ | $77.06 \pm 1.73$ | $78.44 \pm 1.52$ | $79.28 \pm 1.36$ |
| TRIANGLES | $76.02 \pm 2.54$ | $78.44 \pm 1.84$ | $79.91 \pm 1.28$ | $80.27 \pm 1.84$ | $81.74 \pm 2.03$ | $82.58 \pm 1.57$ |

| Dataset | 10-shot | | | 20-shot | | |
|---|---|---|---|---|---|---|
| | No AL | 10 | 20 | No AL | 10 | 20 |
| Reddit | $45.41 \pm 3.79$ | $46.88 \pm 3.14$ | $47.91 \pm 2.99$ | $50.43 \pm 2.66$ | $51.76 \pm 2.32$ | $53.07 \pm 2.21$ |
| ENZYMES | $60.13 \pm 3.98$ | $61.57 \pm 3.48$ | $62.25 \pm 3.06$ | $62.74 \pm 3.64$ | $63.60 \pm 3.30$ | $64.97 \pm 3.11$ |

Thereafter, $l$ *relatively important* samples are chosen from $G_{random}$ and added to $G_N$ for another step of fine-tuning. There can be multiple strategies for defining relative importance of a sample. For our purpose, we define a sample's relative importance via its predicted class probability distribution. We sort these samples in increasing order of the difference between their highest and second highest predicted class probabilities and choose the first $l$ samples from this sorted ranking. We call this importance relative, since each sample is evaluated with respect to the set $G_N$ and thus, there is transductive flow of information among the samples, hence defining the relative embeddings in the space. Intuitively speaking, we have chosen the samples lying closer to separation boundary with respect to $G_N$. The results for various values of $l$ are shown in Table 10. The evaluation is done as mentioned earlier on the unseen set $G_U$. We observe significant improvement for all the datasets. This shows our model is capable of selecting important samples with respect to the few existing samples and learn actively.

## A.7    PERFORMANCE OF MODEL WITH 1 SUPER-CLASS

From table 4 in correspondence to tables 1 and 2, one can observe that the results obtained by using only 1 super-class which is equivalent to removing the super-classes are still better in comparison with many GNN and graph kernel baselines. By removing the super-classes and thus forming the super-graph solely based on k-nearest neighbor heuristic the GAT still learns latent inter class connections via information flow better than the GNNs which use MLP as their classifier. The super graph constructed in such scenario will have arbitrary connections between classes in the beginning, however as the GIN feature extractor learns over time the segregation in the feature space increases leading to better inter as well intra-class connections. Despite this, the performance with super-classes is better as this inductive bias allows the model to initiate with a better alignment in the feature space. The silhouette score comparison for OurMethod-GAT with 1 super-class and with the best performing number of super-classes to GIN and WL clearly indicates the multifold benefits of using GNNs as a classifier via super-graph construction. The t-SNE plots for OurMethod-GAT with only 1 super-class, GIN and WL kernel on the datasets TRIANGLES, Reddit and Letter-High are provided in the figures 7, 8 and 9 respectively.

Table 11: Silhouette coefficients of the test classes for three models - GAT variant of Our Method for 1 super-class which is equivalent to not using any super-classes vs the best performing number of super-classes as well as GIN and WL on 20-shot scenario. For GIN and WL both the sub-columns contain the same values as they don't have any concept of super-classes.

| Method | Reddit-12K | | ENZYMES | | Letter-High | | TRIANGLES | |
|---|---|---|---|---|---|---|---|---|
| | 1-SC | 2-SC | 1-SC | 2-SC | 1-SC | 3-SC | 1-SC | 3-SC |
| GIN | -0.0652 | -0.0652 | 0.0432 | 0.0432 | 0.2316 | 0.2316 | 0.1256 | 0.1256 |
| WL Kernel | -0.0626 | -0.0626 | 0.0366 | 0.0366 | 0.2490 | 0.2490 | 0.0186 | 0.0186 |
| OurMethod-GAT | -0.0593 | -0.0559 | 0.1172 | 0.0989 | 0.3519 | 0.3787 | 0.3975 | 0.4508 |

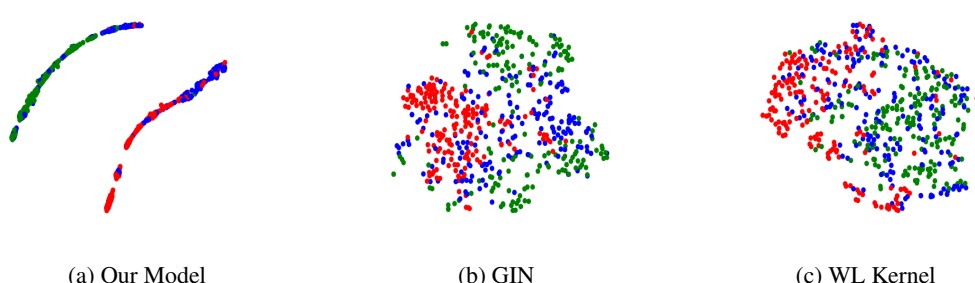

(a) Our Model                    (b) GIN                    (c) WL Kernel

Figure 7: Visualization: t-SNE plots of the computed embeddings of test graphs on 20-shot scenario from OurMethod-GAT with only 1 super-class (left), GIN (middle) and WL Kernel (right) on *TRIANGLES* dataset.

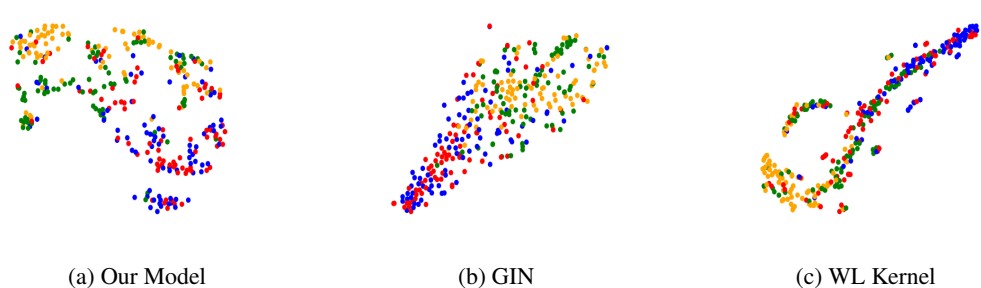

|  |  |  |
|---|---|---|
| (a) Our Model | (b) GIN | (c) WL Kernel |

Figure 8: Visualization: t-SNE plots of the computed embeddings of test graphs on 20-shot scenario from OurMethod-GAT with only 1 super-class (left), GIN (middle) and WL Kernel (right) on *Reddit* dataset.

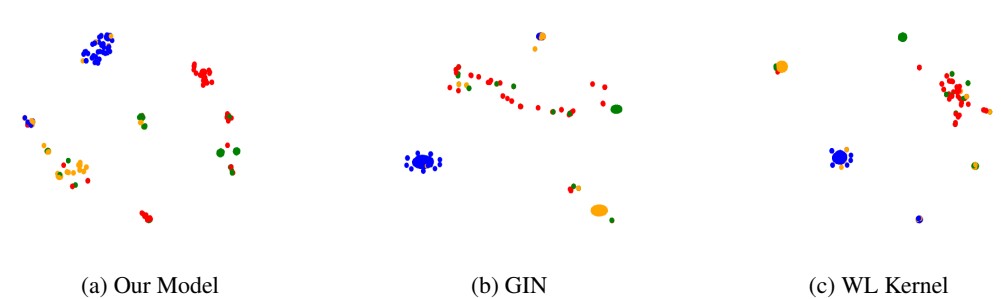

|  |  |  |
|---|---|---|
| (a) Our Model | (b) GIN | (c) WL Kernel |

Figure 9: Visualization: t-SNE plots of the computed embeddings of test graphs on 20-shot scenario from OurMethod-GAT with only 1 super-class (left), GIN (middle) and WL Kernel (right) on *Letter-High* dataset.

