# OpenReview forum: "FEW-SHOT LEARNING ON GRAPHS VIA SUPER-CLASSES BASED ON GRAPH SPECTRAL MEASURES"
_ICLR.cc/2020/Conference — Accept (Poster)_

### Official Review · AnonReviewer3 · 2019-10-23
**Official Blind Review #3**

**Rating:** 6

**Review:**

This paper introduces few-shot graph classification problem and proposes super-class based graph neural network (GNN) to solve it. Experiments on two datasets demonstrate that the proposed model outperforms a number of baseline methods. Some ablation study and analysis are also provided. Followings are my detail review.

It is interesting for the authors to introduce few-shot graph classification problem which is meaningful. If I understood correctly, the authors use graph spectral distance to find prototype graph of each class, then employ prototype graph clustering to obtain super-classes, which are further fed to GNN as the joint optimization of super-class and regular class prediction. To me, the novelty is incremental.

The authors use two new datasets for experiments due to the requirement of numerous class labels. I concerned about performances of different GNN baseline methods in Letter data due to small graph size (with 4.6 nodes in average). The context neighbor information is important for multi-layer GNN. Thus I could not fully judge the effectiveness of proposed model in this data.

In Table 3, I found performance of proposed model with one super-class is still better than different GNN. I did get the point from this result. Why there is no performance decrease as all have the same super-class label? What is the model performance when removing super-class augmentation? I would like to see more discussion or experiment about this.

Update: I am satisfied with author's response and raised my score.

**Experience Assessment:**

I have published one or two papers in this area.

**Review Assessment: Checking Correctness Of Derivations And Theory:**

N/A

**Review Assessment: Checking Correctness Of Experiments:**

I carefully checked the experiments.

**Review Assessment: Thoroughness In Paper Reading:**

I read the paper at least twice and used my best judgement in assessing the paper.

---

> ### Author Response · Authors · 2019-11-09
> **Response to Reviewer 3, Part 1**
>
>  We thank the reviewer for his comments and observations. Following are the answers to each question/comment you have raised.
>
> R3Q1: “It is interesting for the authors to introduce few-shot graph classification problem which is meaningful. If I understood correctly, the authors use graph spectral distance to find prototype graph of each class, then employ prototype graph clustering to obtain super-classes, which are further fed to GNN as the joint optimization of super-class and regular class prediction. To me, the novelty is incremental.”
> R3A1: Yes, your summary can serve as a high-level coarse overview of our work. Regarding the novelty being incremental, we respectfully disagree due to the following reasons.
> The task of few-shot learning on graphs to begin with is novel (as is also pointed out by you and all the other reviewers) and extremely challenging as there are no previous works in this few-shot setting on graphs. The introduction of using the $L_p$-Wasserstein distance between the spectrum of graphs in order to build a supergraph and also cluster class-labels into superclasses is also novel. We are not aware of any graph NNs that have introduced a graph spectral Wasserstein distance between graphs for classification.
>
> We believe our superclass construction per batch embodies the “full context embedding” feature of matching networks [1], which essentially capture a wider context of the support set per batch and not just a single graph and prototype networks [2] by using precomputed Wasserstein prototypes to decide the superclasses that allow information flow between various classes too. Our architecture that jointly learns the superclass and regular class using two-phase (training and fine-tuning phase) training is also a novel construction. Finally, we have also shown our method works well in the semi-supervised and adaptive learning setting (Appendix A.5 and A.6).
> [1] Vinyals et~al. “Matching Networks for One Shot Learning“, NIPS 2016
> [2] Snell et~al. “Prototypical Networks for Few-shot Learning”, NIPS 2017
>
> R3Q2: Low average number of nodes per graph in the datasets used.
> R3A2: Thanks for pointing this out. We found two widely-used graph datasets in graph classification literature with larger average number of nodes, namely Enzymes and Reddit-12K.
> We conducted all our experiments (including sensitivity and ablation studies) on both these datasets and the results have been added to the revised paper. Enzymes has 33 nodes per graph and Reddit-12K has 391 nodes per graph on average.
> Please refer to the table 2 for new results. We find that our results show a marked improvement on the new datasets as well.
> +-------------------------+-----------------------------------------------------------+-----------------------------------------------------------+
> |    Method               |                            Reddit-12K                             |                              Enzymes                               |
> +-------------------------+-------------------+------------------+-------------------+------------------+-------------------+------------------+
> |                                 |       5SHOT     |      10SHOT    |      20SHOT    |       5SHOT     |      10SHOT    |      20SHOT    |
> | WL                          | 40.26 +- 5.17 | 42.57 +- 3.69 | 44.41 +- 3.43 | 55.78 +- 4.72 | 58.47 +- 3.84 | 60.10 +- 3.18 |
> | Graphlet                | 33.76 +- 6.94 | 37.59 +- 4.60 | 41.11 +- 3.71 | 53.17 +- 5.92 | 55.30 +- 3.78 | 56.90 +- 3.79 |
> | AWE                        | 30.24 +- 2.34 | 33.44 +- 2.04 | 36.13 +- 1.89 | 43.75 +- 1.85 | 45.58 +- 2.11 | 49.98 +- 1.54 |
> | Graph2Vec            | 27.85 +- 4.21 | 29.97 +- 3.17 | 32.75 +- 2.02 | 55.88 +- 4.86 | 58.22 +- 4.30 | 62.28 +- 4.14 |
> | Diffpool                 | 35.24 +- 5.69 | 37.43 +- 3.94 | 39.11 +- 3.52 | 45.64 +- 4.56 | 49.64 +- 4.23 | 54.27 +- 3.94 |
> | CapsGNN              | 36.58 +- 4.28 | 39.16 +- 3.73 | 41.27 +- 3.12 | 52.67 +- 5.51 | 55.31 +- 4.23 | 59.34 +- 4.02 |
> | GIN                        | 40.36 +- 4.69 | 43.70 +- 3.98 | 46.28 +- 3.49 | 55.73 +- 5.80 | 58.83 +- 5.32 | 61.12 +- 4.64 |
> | GIN-k-NN              | 41.31 +- 2.84 | 43.58 +- 2.80 | 45.12 +- 2.19 | 57.24 +- 7.06 | 59.34 +- 5.24 | 60.49 +- 3.48 |
> | OurMethod-GCN | 40.77 +- 4.32 | 44.28 +- 3.86 | 48.67 +- 4.22 | 54.34 +- 5.64 | 58.16 +- 4.39 | 60.86 +- 3.74 |
> | OurMethod-GIN  | 41.59 +- 4.12 | 45.67 +- 3.68 | 50.34 +- 2.71 | 55.42 +- 5.74 | 60.64 +- 3.84 | 62.81 +- 3.56 |
> +-------------------------+-------------------+------------------+-------------------+------------------+-------------------+------------------+

---

> > ### Author Response · Authors · 2019-11-09
> > **Response to Reviewer 3, Part 2**
> >
> > R3Q3: Why there is no performance decrease as all have the same superclass label?
> > R3A3: While all the graphs possess the same superclass label (SC), there is still a supergraph (k-NN graph) constructed on ALL the graph feature vectors belonging to the single SC, which captures inter-class information flow. The only difference is that with more superclasses, there are separate k-NN graphs for each superclass and there is no unnecessary information flow between classes that don’t belong to the same superclass, whereas in the 1-SC situation, we end up relaxing this restriction and have just a single k-NN graph across all class labels grouped under the single SC. This supergraph is then fed to $C^{GAT}$ which still learns and manages to further increase class separation that was present in the feature vectors extracted by GIN. Therefore, even with a single SC our $C^{GAT}$ manages to outperform other baseline GNNs.
> >
> > We further generated t-SNE plots and a silhouette coefficient table (refer to Appendix section 7) to study the class-separation effect of “just a GIN”, versus our method which uses a “GIN followed by a GAT with 1 SC” and other best performing number of superclasses for each dataset. We observe that our method with 1 SC outperforms both GIN and WL-kernel.
> > SILHOUETTE COEFFICIENT TABLE
> > +-------------------------+-------------------+-----------------+-----------------------+--------------------------+--------+
> > |    Method              |     Reddit-12K        |    ENZYMES       |    Letter-High     |    TRIANGLES    |
> > +-------------------------+-------------------+------------------+-------------------+------------------+-------------------+
> > |                                 | 1-SC     | 2-SC       |  1-SC   |  2-SC    |  1-SC   |   3-SC   |   1-SC   |   3-SC   |
> > | GIN                         | -0.0652 | -0.0652 | 0.0432 | 0.0432 | 0.2316 | 0.2316 | 0.1256 | 0.1256 |
> > | WL Kernel              | -0.0626 | -0.0626 | 0.0366 | 0.0366 | 0.2490 | 0.2490 | 0.0186 | 0.0186 |
> > | OurMethod-GAT  | -0.0593 | -0.0559 | 0.1172 | 0.0989 | 0.3519 | 0.3787 | 0.3975 | 0.4508 |
> > +-------------------------+-------------------+------------------+-------------------+------------------+-------------------+
> >
> > R3Q4: What is the model performance when removing super-class augmentation?
> > R3A4: We refer you to the ablative studies conducted by us in Table 3 of the main paper which shows the comparison between “No super class” and “with super class”.

---

### Official Review · AnonReviewer1 · 2019-10-29
**Official Blind Review #1**

**Rating:** 3

**Review:**

This paper proposed a few-shot graph classification algorithm based on graph neural networks. The learning is based on a large set of base class labeled graphs and a small set of novel class labeled graphs. The goal is to learn a classification algorithm over the novel class based on the sample from the base class and novel class. The learning process constitutes of the following steps. First, the base class is classified into K super classes based on the spectral embedding of the graph (onto distributions over the corresponding graph spectrum) and the k-means algorithm with the Wasserstein metric. Second, for each super class, the classification is done through a feature extractor and a classifier. In the training of the feature extractor and classifier, the author introduces a super-graph with each node representing a super class. Finally, in the fine-tuning stage, the feature extractor is fixed, and the classifier is trained based on the novel class.

This work seems to be the first attempt to adopt the few-shot learning in graph classification tasks. The architecture is novel, and the classification of graph based on spectral embedding together with the Wasserstein metric is novel to me.

I vote for rejecting this submission for the following concerns.

1. The classification of base class into super classes seems questionable to me. In the meta-learning language, the author attempts to learn a good representation of graphs based on different graph classification tasks generated by a task distribution. In terms of graph classification, the task distribution is supported on the joint distributions (G, Y). Hence, to characterize different tasks, as far as I am concerned, the classification should take both the graph G and the label Y into consideration, instead of solely the graph.

2. Though seemingly very important to the architecture, the purpose of constructing the super-graph g^{sup} in the training of C^{CAT} seems to be unclear to me.  I would appreciate it if the author could provide more explanation on the introduction of the super-graph in training.


**Experience Assessment:**

I do not know much about this area.

**Review Assessment: Checking Correctness Of Derivations And Theory:**

I assessed the sensibility of the derivations and theory.

**Review Assessment: Checking Correctness Of Experiments:**

I assessed the sensibility of the experiments.

**Review Assessment: Thoroughness In Paper Reading:**

I made a quick assessment of this paper.

---

> ### Author Response · Authors · 2019-11-09
> **Response to Reviewer 1**
>
> We thank the reviewer for his comments and observations. Following are the answers to each question you have raised.
> R1Q1(a): “The classification of base class into super classes seems questionable to me. In the meta-learning language, the author attempts to learn a good representation of graphs based on different graph classification tasks generated by a task distribution. In terms of graph classification, the task distribution is supported on the joint distributions (G, Y).”
> R1A1(a): Our model has two major components, $C^{sup}$ (for super-class prediction) and $C^{GAT}$ (for graph label prediction). During the training phase of our classification, $C^{sup}$, which is a MLP layer, learns the super-class labels of the samples based on GIN’s extracted feature vectors (which represent base class labeled graphs). While, $C^{GAT}$ takes as input the “graph of graphs” (supergraph) which models the latent inter-class as well as intra-class information and is constructed in every training batch, along with base-class labels, to learn the associated class distribution.
> Then, during the fine-tuning phase on graphs with novel class labels, the feature extractor’s (GIN) parameters are fixed and $C^{sup}$ is used to infer the super-class label of the novel class labeled graphs. Then, the parameters learned by $C^{GAT}$ get updated and further “fine-tuned” for better performance on the novel samples.
> In addition to our brief overview, you could also find a very neatly detailed summarization of our method in reviewer 2’s comments (paragraphs 1-4).
>
> The meta-learning framework, where batches are sampled as “episodes” with N-way K-shot setting, does not perform as well in our few-shot setting on graphs for the following reasons:
> 1) We have very limited total number of training classes (in order of 10s), when compared to the image domain (order of 100s and 1000s). This limitation hampers learning across tasks and generalization to new unseen tasks.
> 2) In each of our batches, we randomly sample a fixed-size of training samples belonging to the set of N labels chosen. Therefore, when building our supergraph, we end up with k-NN graphs of “variable size” per super-class, compared to fixed size (K nodes) k-NN graphs that we would have got using episodic learning. We suspect this further allows our GAT to learn and generalize better to unseen graphs.
> Furthermore, in [1], the authors use a similar strategy in their “baseline++” method and produce good results. Their findings are also in sync with our empirical finding.
> [1] Chen et~al. “A closer look at Few-Shot Classification”, ICLR 2019
>
> R1Q1(b): Hence, to characterize different tasks, as far as I am concerned, the classification should take both the graph G and the label Y into consideration, instead of solely the graph.”
> R1A1(b): In every batch of graphs during both training and fine-tuning phase, each graph is associated with its corresponding graph label. In case of training, its a base-class and in the case of fine-tuning its a novel class. In the case of $C^{GAT}$, the graph is accompanied by a regular class label and in case of $C^{sup}$, the graph is accompanied by a superclass label.
>
> R1Q2: “Though seemingly very important to the architecture, the purpose of constructing the super-graph $g^{sup}$ in the training of $C^{CAT}$ seems to be unclear to me.”
> R1A2: What makes few-shot learning particularly difficult compared to common machine learning settings is the dearth of training examples, which results in a bad empirical risk approximation for the expected risk and therefore gives rise to an empirical risk minimizer that is sub-optimal. Reducing the required sample complexity can result in a better empirical risk minimizer. Therefore, given a very large space of hypotheses H, our goal is to further restrict and constrain H using some prior knowledge because a reduced H has reduced sample complexity and thus requires fewer training samples to be trained. We provide this “prior knowledge” in the form of a “graph of graphs”, namely our super-graph $g^{sup}$, which captures both the latent inter-class and intra-class relationships between classes. Observe that in $g^{sup}$, we build a k-NN graph PER super-class, restricting any flow of information between super-classes, thus further restricting H. We force our model to jointly learn both the superclass and graph class labels. This way similar classes (grouped under a superclass) together contribute to learning a general prior representing the superclasses and each superclass also provides “guidance” to better train with the few samples assigned to that superclass.
> The introduction of this prior knowledge in the form of a supergraph in $C^{GAT}$ during training also helps generalize better to the novel samples that are presented to our model in the fine-tuning stage.
>
> Additionally, we would also like to draw attention to the supergraph usage summary provided by reviewer 2 in paragraph 3 of their comments.

---

### Official Review · AnonReviewer2 · 2019-11-03
**Official Blind Review #2**

**Rating:** 6

**Review:**

This paper introduces few-shot learning for graph classification. The authors propose a pre-training->fine-tuning approach to handle graph classes unseen at training time (and in only a few shots at test time).

At a high level, and to my understanding, their method a priori generates the ingredients for a graph of graphs, a "super-graph", in two steps: first, it discovers a prototype graph for each graph class in the dataset, then it clusters the prototype graphs into a set of super-classes by k-NN. Both of these operations rely on the spectral distance between graphs, defined in this work using the spectrum of a graph's normalized Laplacian matrix and the pth Wasserstein distance between probability measures. The intuition is that the super-graph built from these ingredients helps model latent relations between graph classes; these relations can be used at test time to improve classification of unseen graph classes.

During (pre-)training, the model builds super-graphs on each batch of data. It uses super-graph information in two ways: an auxiliary classification head (an MLP called C^sup) is trained to map graph embeddings to their corresponding super-class labels, and the super-graph itself, whose nodes are embeddings for individual graphs in the batch, passes through a graph attention network (GAT) that outputs a base class for each graph -- this is the classification head C^GAT. The graph embeddings themselves come from a feature extractor F_θ implemented as a graph isomorphism network (GIN).

During the fine-tuning stage the model adapts to and classifies graphs from classes unseen during training. Here the parameters of the feature extractor GIN and the C^sup MLP are frozen. C^sup outputs a set of super-class labels that are used to construct a super-graph, which in turn feeds into C^GAT, which in turn yields labels for the test graphs. C^GAT (but not the GIN or C^sup) fine-tunes on a small number q of labelled examples of each novel test class. The full model is evaluated on unlabelled examples from the novel test classes. This process assumes that the novel test graph classes belong to the same set of super classes as the training graph classes, a point that is, unfortunately, not discussed.

There's a lot to digest in this paper, on both the technical and architectural sides. There are graphs of graphs (super-graphs) and different GNN variants operating on both, with the output of one graph network, the feature extracting GIN, feeding into another, the GAT classifier. Understanding all of these pieces and how they fit together is challenging for the reader: I got lost somewhat in the Classifier description in Section 4, while Section 3 defines many things and gives some math that might be extraneous. It is also not immediately obvious that fine-tuning takes place on the set G_N and testing on the set G_U. Overall, though, the paper became clear to me with time and I found the overall presentation to be good. Some additional figures that depict the super-graph construction and clustering would be useful.

The construction and use of the super-graph structure to model relations between graph classes is interesting and novel to me, though it relies on well-established techniques (Wasserstein barycenters, Lloyd's algorithm for k-NN). The architecture itself, which combines GINs and GATs, is also novel to me; a downside is that it is highly complex.

Experiments were undertaken on two datasets and seem fairly thorough, with variance established on a high number of seeds (high in the deep learning literature). They demonstrate that the proposed method makes significant improvements over baselines. The baselines are somewhat limited because, as the authors state, "there do not exist any standard state-of-the-art methods for few-shot graph classification". However, I do not think the authors should be penalized for trying something new. On the other hand, given the novelty of the task, it would be nice to see an investigation/discussion of how few-shot graph learning differs from few-shot image learning (where there has been much more work).

I found the ablation and sensitivity studies illuminating, and I was pleased to see that the authors do support their claim that the super-graphs improve class separation over the feature extractor embeddings -- the GIN-k-NN baseline results provide evidence of this.

One place where I lack confidence in the results: I am not very familiar with the datasets used (TRIANGLES and Letter-High) nor how standard they are in the graph-learning literature. The authors do not even describe in the paper what the graph classes in these datasets actually are or represent, which would be good to know.

Overall, I think the paper is worth seeing and discussing at the conference, although it could be improved in various ways.

Minor errors:
- there appears to be bracket imbalance in eq.6
- "Lloyd's" is misspelled a few times

Reviewer's note: I have significant experience in few-shot learning but not in graph neural networks.

**Experience Assessment:**

I have published one or two papers in this area.

**Review Assessment: Checking Correctness Of Derivations And Theory:**

N/A

**Review Assessment: Checking Correctness Of Experiments:**

I assessed the sensibility of the experiments.

**Review Assessment: Thoroughness In Paper Reading:**

I read the paper at least twice and used my best judgement in assessing the paper.

---

> ### Author Response · Authors · 2019-11-09
> **Response to Reviewer 2, Part 1**
>
> We thank the reviewer for his comments and observations. We are in total agreement with the detailed overview of our method provided by you. Following are the answers to each question/comment you have raised.
>
> R2Q1: “This process assumes that the novel test graph classes belong to the same set of super classes as the training graph classes, a point that is, unfortunately, not discussed.”
> R2A1: Thanks for pointing out the omission of this crucial discussion. Yes, we assume that the novel test classes belong to the same set of superclasses from the training graphs. The reason being that the novel class labeled samples are so much fewer than the base class labeled samples that the resulting supergraph ends up being extremely sparse and deviates a lot from the shape of the supergraph from the base classes; therefore it severely hinders $C^{GAT}$’s ability to effectively aggregate information from the embeddings of the novel class labeled graphs. Instead, we pass the novel graph samples through our trained $C^{sup}$ and infer its super-class label and this works very effectively for us, as is evidenced by our empirical results. We have accordingly updated our draft and this explanation can be found in the new Discussion subsection of Section 4.2 of our revised draft.
>
> R2Q2: I got lost somewhat in the Classifier description in Section 4, while Section 3 defines many things...
> R2A2: We have updated the description in Section 4 by adding in the new discussion section concerned above.
>
> R2Q3: It is also not immediately obvious that fine-tuning takes place on the set $G_N$ and testing on the set $G_U$.
> R2A3: We have described this in the “problem definition” subsection in Section 3 (Preliminaries). Additionally, we added the sentence “Finally the evaluation is performed on the samples from the unseen test set $G_U$.” at the end of our Classification subsection of 4.2.
>
> R2Q4: Some additional figures that depict the super-graph construction and clustering would be useful.
> R2A4: Thanks. We have added an illustration (Fig 2) that depicts the working of our Wasserstein super-class k-means clustering algorithm.

---

> > ### Author Response · Authors · 2019-11-09
> > **Response to Reviewer 2, Part 2**
> >
> > R2Q5: Discussion of few-shot in graphs vs few-shot in images.
> > R2A5: Few shot learning (FSL) has gained wide-spread traction in the image domain in recent years. However the success of FSL in images is not easily translated to the graph domain for the following reasons:
> >       (a) Images are typically represented in Euclidean space and thus can easily be manipulated and
> >             handled using well-known metrics like cosine similarity, $l_p$ norms etc. However, graphs originate
> >             from non-Euclidean domains and exhibit much more complex relationships and interdependency
> >             between objects. Furthermore, the notion of a distance between graphs is also not
> >             straightforward and requires construction of graph kernels or the use of standard metrics on
> >             graph embeddings [1]. Additionally, such graph kernels don’t capture higher order relations very
> >             well.
> >       (b) In the FSL setting on images, the number of training samples from various classes is also
> >             abundantly  more than what is available for graph datasets. The image domain allows training
> >             generative models  to learn the task distribution and can further be used to generate samples for
> >             “data augmentation”, which act as very good priors. In contrast, graph generative models are still
> >             in their infancy and work in very restricted settings [2] . Furthermore, methods like cropping and
> >             rotation to improve the models can't be used for graphs given the permutation invariant nature
> >             of graphs. Additionally, removal of any component from the graph can adversely affect its
> >             structural properties, such as in biological datasets.
> >       (c) The image domain has very well-known regularization methods (e.g. Tikhonov, Lasso)  that help
> >             generalize much better to novel datasets. Although, they don’t bring any extra supervised
> >             information and hence cannot fully address the problem of FSL in the image domain. To the best
> >             of our knowledge, this is still an open research problem even in the image domain. On the other
> >             hand,  in the graph domain, our work would be a  first step towards graph classification in an FSL
> >             setting, which would then hopefully pave the path for better FSL graph regularizers.
> >      (d) Transfer learning [3] has led to substantial improvements on various image related tasks due to
> >             the  high degree of transferability of feature extractors. Thus, downstream tasks like few-shot
> >             learning can  be  performed well with high quality feature extractor models, such as Resnet
> >             variants trained on Imagenet. Transfer learning, or for that matter even good feature extractors,
> >             remains a daunting challenge in the graph domain. For graphs, there neither exists a dataset
> >             which can serve as a pivot for high quality feature learning, nor does there exist a Graph NN which can capture the higher order relations between various categories of graphs, thus making this a highly challenging problem.
> > [1]  Kriege et~al. “A survey on graph kernels”, Arxiv 2019
> > [2]  Simonovsky et~al, “GraphVAE: Towards Generation of Small Graphs Using Variational Autoencoders”, Arxiv 2018
> > [3]  Tan et~al. “A Survey on Deep Transfer Learning”, Arxiv 2018
> >
> > R2Q6: The authors do not even describe in the paper what the graph classes in these datasets actually are or represent, which would be good to know.
> > R2A6: Experiments on two new datasets, Enzymes and Reddit-12K have been added to the paper. These two datasets are popular and have been extensively used in graph classification literature prior to this work. We now describe each dataset.
> >     (a) Reddit-12K - dataset contains 11929 graphs where each graph corresponds to a thread in which
> >           each node represents a user and each edge represents that one user has responded to a comment
> >           from some other user. There are 11 different types of discussion forums corresponding to each of the
> >           11 classes.
> >    (b) ENZYMES - is a dataset of protein tertiary structures consisting of 600 enzymes from the BRENDA
> >          enzyme database. The dataset contains 6 different graph categories corresponding to each different
> >          top-level EC enzyme.
> >    (c) TRIANGLES- dataset contains 10 different classes where the classes are numbered from 1 to 10
> >          corresponding to the number of triangles/3-cliques in each graph of the dataset.
> >    (d) Letter-High - dataset contains graphs which represent distorted letter drawings from the english
> >          alphabet - (A, E, F, H, I, K, L, M, N, T, V, W, X, Y, Z). Each graph is a prototype manual construction of the
> >          alphabets.
> >
> > R2Q7:
> > Minor errors:
> > - there appears to be bracket imbalance in eq.6
> > - "Lloyd's" is misspelled a few times
> > R2A7: Thanks for pointing out these errors. We have corrected them in the revised draft.

---

### Author Response · Authors · 2019-11-09
**Response to All Reviewers**

Dear Reviewers,
We would like to thank all of you for your questions, comments, and suggestions.
Along with our rebuttal, we have also updated the draft with new experiments on
two additional datasets. Below is a list of the changes we have made to the
revised draft along with their rebuttal answer IDs for easy cross-referencing.
1) (R2A5) Comparison of FSL in graph versus image domain:Added after Related Work.
2) (R2A4) Figure 2 providing description of our Wasserstein super-class k-means clustering algorithm added in main paper.
3) (R2A1) Discussion of Super Class Construction of Novel classes: Last paragraph of section 4.2
4) (R2A3) "Finally the evaluation is performed on the samples from the unseen test set $G_U$" added at the end of the
     Classifier subsection in Section 4.2
5) (R2A6) Section 5.2 Updated with ENZYMES and REDDIT-12K datasets along with experimental results in Table 2.
6) (R3A2) Sections 5.3, 5.4 Updated along with their tables for ENZYMES and REDDIT-12K datasets.
7)  Section A.1 updated with descriptions of the datasets.
8) (R3A2) Sections A.4, A.5 and A.6. Updated their tables for silhouette scores, semi-supervised and active learning settings
      on the new datasets.
9) (R3A3) Section A.7 added with new t-SNE plots and tables for silhouette coefficients.

---

### Decision · Program_Chairs · 2019-12-19

**Decision:**

Accept (Poster)

**Comment:**

The authors propose a method for few-shot learning for graph classification. The majority of reviewers agree on the novelty of the proposed method and that the problem is interesting. The authors have addressed all major concerns.